# MECHANISTIC ANALYSIS OF DEMONSTRATION CONFLICTS IN IN-CONTEXT LEARNING

## ABSTRACT

In-context learning (ICL) enables large language models (LLMs) to perform novel tasks through few-shot demonstrations. However, demonstrations *per se* can naturally contain noise and conflicting examples, making this capability vulnerable. To understand how models process such conflicts, we study demonstration-dependent tasks where models must infer underlying patterns—scenarios we characterize as rule inference. We find that models suffer substantial performance degradation from single corrupted demonstrations, with corrupted rules accounting for 80% of wrong predictions despite appearing in only one position. This systematic misleading behavior motivates our investigation of how models process conflicting evidence internally. Using linear probes and logit lens analysis, we discover models encode both correct and incorrect rules in early layers but develop prediction confidence only in late layers, revealing a two-phase computational structure. We identify attention heads for each phase underlying the reasoning failures: Vulnerability Heads in early-to-middle layers exhibit positional attention bias with high sensitivity to corruption, while Susceptible Heads in late layers significantly reduce support for correct predictions when exposed to single corrupted evidence. Targeted ablation validates our findings, with masking identified heads improving performance by over 10%. Our work establishes a mechanistic foundation for understanding how LLMs process conflicting evidence during in-context rule learning. Our code is available at `https://anonymous.4open.science/r/icl-demo-conflict-3E47/`.

## 1 INTRODUCTION

In-context learning (ICL) enables large language models (LLMs) to adapt to new tasks through few-shot demonstrations without parameter updates (Brown et al., 2020). A particularly intriguing aspect of ICL is the capability to infer underlying patterns from demonstrations, marking a form of general intelligence akin to human reasoning (Dong et al., 2022; Bai et al., 2023; Kossen et al., 2023). For example, LLMs are able to learn and perform competitively on linear regression and analogical reasoning problems, demonstrating their capacity to learn abstract patterns from limited contextual information (Von Oswald et al., 2023; Coda-Forno et al., 2023; Zhang et al., 2024).

However, learning the novel rules in-context can be intrinsically vulnerable to conflicting demonstrations, with ICL performance broadly known to degrade substantially when demonstrations contain contradictory information (Yoo et al., 2022; Min et al., 2022; Wei et al., 2023). This vulnerability is particularly concerning given that real-world data naturally contains noise and outliers that can mislead models' judgment about underlying patterns. While previous work has examined the conflict resolution in general tasks such as text classification and question answering (Li et al., 2023; Xie et al., 2023; Halawi et al., 2023; Yu & Ananiadou, 2024), these findings cannot be directly translated to rule inference settings. Notably, unlike the tasks where models already possess sufficient parametric knowledge [1] for reliable zero-shot performance (Yang et al., 2022; Jiao et al., 2023), rule inference requires genuine demonstration reliance.

In this work, we first investigate the dynamics of in-context rule inference under conflicting demonstrations. We employ two Operator Induction tasks (Zong et al., 2024) (Figure 1 left) and Fake Word

---

[1] known as the factual and relational information that an LLM acquires after pre-training.

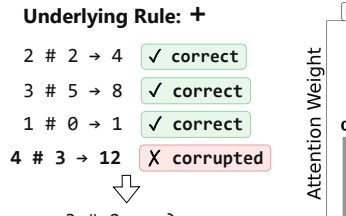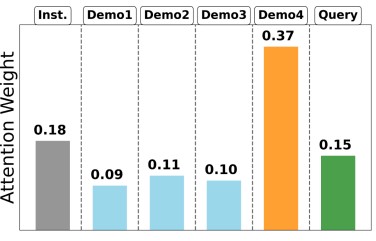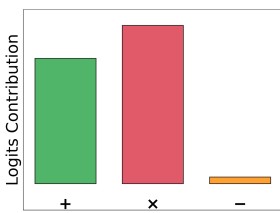

Figure 1: **Left:** Operator induction task where models are required to infer the underlying rule (+) from demonstrations, among which we introduce one corrupted example. **Middle:** Vulnerability heads show disproportionate attention allocation; they also show drastic output changes when the heavily-attended position corrupts. **Right:** Susceptible heads shows logit contributions favoring the corrupted operator (×) over the correct one (+) despite minority corruption.

Inference (mapping synthetic vocabulary to real concepts), where models perform well with ICL but can only achieve chance-level performance without examples. In these tasks, the modularity of demonstrations allows us to introduce conflicts by corrupting rules at specific positions without undermining others. Our experiments reveal that LLMs suffer substantial performance degradation from a single corrupted example among a vast majority of correct ones. We also observe systematic performance variance when corrupting different positions of demonstrations, also known as the positional bias (Wang et al., 2024; Cobbina & Zhou, 2025)

During evaluation, we notice a strong signal that models are misled by the injected conflict—when corruption flips models from correct to wrong, the corrupted operator accounts for 80% of wrong predictions despite appearing in only one position. Given this, we speculate that models encode both rules during the process, yet can fail to resolve the correct rule at the end. This drives us to study how conflicting evidence evolves during model's internal processing and further identify model components that cause and resolve them.

To study this, we first train linear probes (Alain & Bengio, 2016) to track the emergence of each operator in the processing, showing that the models encode both correct and misleading rule in early and intermediate layers. Then, we run logit lens (nostalgebraist, 2020) to decode model predictions layer by layer, revealing that models develop strong confidence for both rules at very late layers. This temporal separation suggests a natural two-phase hypothesis: conflict creation and conflict resolution.

To test this hypothesis, we seek to identify model components responsible for each phase. For the first phase, since demonstrations are independent and equivalent in our tasks, the positional bias indicates that specific components treat positions unequally. Building on prior work that attributes this bias to attention heads (Wang et al., 2024), we localize *vulnerability heads* (Figure 1 middle) that both attend disproportionately to certain positions and show high sensitivity when those positions are corrupted, finding their concentration in early-to-middle layers. For the second phase, we localize *susceptible heads* (Figure 1 right) whose contribution to predicting wrong operators increases the most when the corruption is introduced to clean demonstrations, finding their concentration in later layers. Ablating these two types of heads mitigates the performance degradation to more than 10%, confirming the causal roles of both head types in creating and falsely resolving conflicts. Finally, our correlational test reveals a synergy between two heads: masking vulnerability heads more significantly reduces the contribution of false resolution heads towards the corrupted operator.

Our contributions are twofold. First, we establish a foundational framework for studying the dynamics of conflict resolution during in-context rule inference. Second, we provide mechanistic evidence for a two-phase structure of reasoning under corruption, identifying specific model components that exhibit both correlational and causal relevant to vulnerability creation and mitigation.

## 2 RELATED WORK

### 2.1 CONFLICT RESOLUTION IN IN-CONTEXT LEARNING

Our work studies scenarios where LLMs encounter conflicts during in-context learning. Research indicates two types of conflicts emerging in ICL (Xu et al., 2024): context-memory conflicts where

contextual information contradicts the model's parametric knowledge, and inter-context conflicts where multiple demonstrations contradict. While both conflicts are common in real world (Xu et al., 2024), prior investigations predominantly focus on context-memory conflicts, examining how models fail to resolve collisions between their encoded knowledge and contextual information, leading to wrong predictions in tasks like text classification, textual and visual question answering, etc. (Pan et al., 2021; Min et al., 2022; Xie et al., 2023; Hua et al., 2025; Halawi et al., 2023) In contrast, we focus on inter-context conflicts in in-context rule-learning scenarios. While models demonstrate strong capabilities on rule inference tasks like operator induction, linear regression, and analogical reasoning (Von Oswald et al., 2023; Coda-Forno et al., 2023; Zong et al., 2024), how conflicting demonstrations affect their reasoning dynamics remains underexplored.

## 2.2 MECHANISTIC INTERPRETABILITY OF IN-CONTEXT LEARNING

Our approach to understanding conflict processing mechanisms follows the work of mechanistic interpretability, which seeks to reverse-engineer the computational framework underlying neural network behaviors (Olah et al., 2020; Elhage et al., 2021; Cammarata et al., 2021). These methods enable us to identify when and where models encode specific information, trace how representations evolve across layers, and localize model components responsible for particular behaviors (Wang et al., 2022; Meng et al., 2022; Jiao et al., 2023; Gurnee et al., 2023).

Recent work applying mechanistic interpretability to ICL has revealed foundational insights about demonstration processing. Notably, Olsson et al. (2022) discovered induction heads, the attention mechanism crucial for pattern completion and copying behaviors in ICL. Studies have identified specific attention heads responsible for various ICL phenomena: mechanisms underlying few-shot learning capabilities (Wang et al., 2022), heads that promote incorrect predictions under adversarial contexts (Halawi et al., 2023), and components that contribute to positional bias (Han et al., 2024). This body of work establishes attention heads as primary computational substrates for ICL processing, providing the foundation for our component-level analysis of conflict creation and conflict resolution.

## 3 DYNAMICS IN RULE INFERENCE UNDER CONFLICTING DEMONSTRATIONS

We start by formulating the task of ICL that we study: LLMs relying on few-shot demonstrations to infer the underlying rule of a novel task. To observe model's reasoning dynamics under conflict, we establish an intervention framework that introduces targeted conflicts among demonstrations.

### 3.1 A CORRUPTION-BASED INTERVENTION FRAMEWORK

**Task Setup.** Without loss of generality, we formulate the task as where language models are instructed to infer latent rules from demonstrations $\mathcal{D} = \{(x_i, y_i)\}_{i=1}^{k}$ and apply the learned rule to predict $\hat{y}_q = r^*(x_q)$ for query $x_q$. Clean demonstrations represent unanimous evidence for the underlying rule $r^* \in \mathcal{R}$, where $\mathcal{R}$ is the hypothesis space of possible rules. To introduce conflicting evidence among demonstrations, we intervene with a position-specific corruption at demonstration index $p \in \{1, \ldots, k\}$, by replacing the correct output $y_p = r^*(x_p)$ with $y_p^{\text{corrupt}} = r'(x_p)$ where $r' \neq r^*$. The model is then required to do the exact same task under the corrupted demonstration set as:

$$\mathcal{D}_p^{\text{corrupt}} = \{(x_1, y_1), \ldots, (x_{p-1}, y_{p-1}), (x_p, y_p^{\text{corrupt}}), (x_{p+1}, y_{p+1}), \ldots, (x_k, y_k)\} \tag{1}$$

This setup carries three fundamental principles. First, we require **demonstration reliance**, where models at most exhibit chance-level performance in zero-shot settings, ensuring their genuine reliance on demonstrations rather than on their parametric knowledge. Secondly, during the evaluation of performance under corruptions, we impose a **majority rule** by introducing conflicting demonstrations only when correct ones substantially outnumbers corrupted ones. Thirdly, we require the **modularity** of demonstrations during inference, where each demonstration is independent and provides equally weighted evidence of the underlying rule. Collectively, these three principles provide a valid foundation for studying reasoning dynamics through which LLMs inference rules with contradictions.

**Datasets and Models.** Note that a wide variety of ICL tasks either involve interdependent demonstrations or potentially provide unequal weights per demonstration (Wei et al., 2021), we carefully adopt

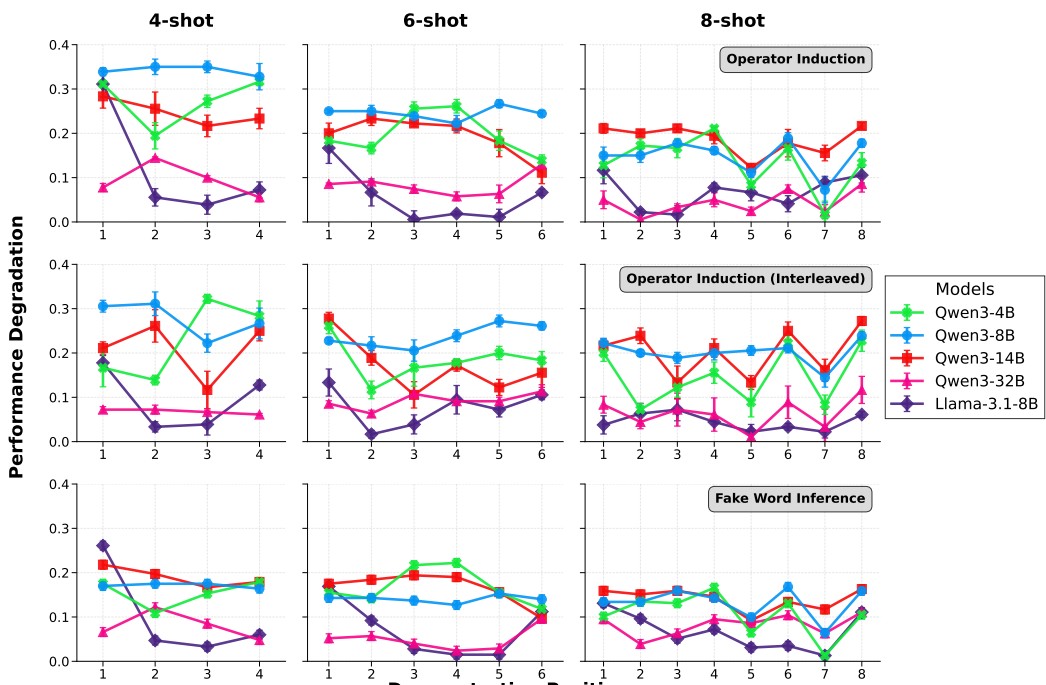

Figure 2: Performance degradation under single-position corruption across five language models and three tasks. Each point represents the decrease in accuracy when corrupting the demonstration at that specific position, relative to the performance when given warning instructions.

three tasks that adhere to our principled design criteria: Operator Induction and Interleaved Operator Induction (Zong et al., 2024), where models must infer the underlying mathematical operation from demonstrations to answer the final query, and Fake Word Inference, where models learn mappings from synthetic vocabulary to real concepts (e.g., "blimontar glemorax" → "red hat"). The Fake Word task naturally prevents parametric knowledge reliance, as models cannot have encountered these nonsense words during pretraining, while maintaining demonstration modularity through independent word-concept pairs. We conduct experiments across five LLMs, including the Qwen3 collection (Qwen-3-4B-Instruct, Qwen3-8B, Qwen3-14B, and Qwen3-32B) (Yang et al., 2025) and Llama-3.1 models (Llama-3.1-8B-Instruct) (Dubey et al., 2024).

**Evaluation Protocol.** Given our focus on performance dynamics, we are interested in how often models maintain correct predictions despite shown with conflicts. However, perturbations of demonstrations can inherently disturb model predictions heavily (Floridi & Chiriatti, 2020; Wei et al., 2021). To calibrate this variability, we compare standard ICL performance against controlled performance when models receive explicit warnings about potential demonstration errors and instructions to apply majority reasoning. Given the majority rule of our framework, we interpret the performance gap between warned and unwarned conditions as reflecting the model's difficulty in autonomously detecting and resolving conflicts. Details of our prompts, instructions, and the evaluation pipeline are included in Appendix B.

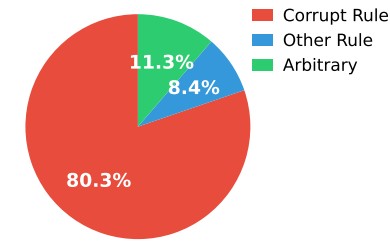

Figure 3: Corrupted rules predict 80% of answers when models flipped from right to wrong under corruption.

## 3.2 Consistent Performance Degradation with Single-Position Corruption

Following our corruption framework, we systematically evaluate model performance with different numbers of demonstrations across all tasks. We first validate models' in-context learning capability over tasks in Table 2 at Appendix B. To observe performance dynamics under conflicts, we run position-specific corruption by corrupting only a single demonstration at each position within the

demonstration sequence and examine the calibrated performance gap position by position, maintaining the majority rule.

As shown in Figure 2, LLMs exhibit consistent and substantial performance degradation under conflicts. The degradation can reach as high as 0.35 when Qwen3-8B performs the 4-shot Operator Induction task, with an average degradation of 0.15 across all cases. Notably, this degradation persists across model scales and architectures, even though all selected LLMs demonstrate reliable ICL capability in our test cases. Moreover, substantial performance degradation persists even as the number of demonstrations increases to 8, where correct samples maintain a dominant 7:1 advantage over conflicting ones. While we observe a modest improvement with additional demonstrations, this effect is far from sufficient to restore performance to the calibrated baseline. This pattern indicates that current models are significantly affected by wrong information during in-context rule inference.

**Models Systematically Adopt Corrupted Rules.** To understand how models fail under corruption, we analyze the pattern of wrong predictions when models flip from correct to incorrect answers. By examining which rule underlies each wrong prediction, we can determine whether models are systematically misled by corrupted evidence or failing randomly. As shown in Figure 3, we note that the corrupted operator accounts for 80.3% of these wrong answers in the Operator Induction task, despite appearing in only one demonstration position. This systematic adoption of minority corrupted evidence, rather than arbitrary errors, suggests that models actively encode and process conflicting rules during inference, yet fail to properly resolve which rule to apply for final predictions.

**Positional Bias Emerges from Corruption Patterns.** Moreover, our experimental design reveals notable positional bias (Hsieh et al., 2024; Cobbina & Zhou, 2025) as shown in Figure 2, where identical corruptions produce different performance impacts depending on their location within the demonstration sequence. This observed bias is particularly instructive in our framework, given the independence and equivalence nature of demonstrations. Under ideal conflict resolution, models should exhibit uniform vulnerability to corruption across all positions, yielding near-zero position bias. The authentic nature of this position bias, uncoupled from confounds of demonstration-specific and parametric knowledge reliance, provides a clean signal for our analysis.

## 4    INTERNAL DYNAMICS OF RULE INFERENCE ICL UNDER CORRUPTION

In the previous section, we present evaluation experiments that reveal clear LLM performance degradation under corruption. However, these evaluations provide little insight into when and how LLMs internally represent and resolve conflicting demonstrations during in-context learning. In this section, we aim to understand these questions by analyzing model internal representations.

### 4.1    LLMS ENCODE UNDERLYING RULES IN EARLY-TO-MIDDLE LAYERS

We use linear probes, trained linear classifiers for detecting the presence of specific information in model internal representations (Alain & Bengio, 2016), to identify where models encode the rule information. To investigate the evolution of rule encoding, we train linear probes to identify each possible underlying rule in the residual stream across all transformer layers. Details of the probe training setup and validation can be found in Appendix C.

We then evaluate model's encoded confidence of detecting the rules from demonstrations, measured by the predictions of linear probes, under two corruption scenarios: no corruption (all demonstrations are correct) and minority corruption (correct demonstrations outnumber corrupted ones). As shown in Figure 4, probing results reveal two patterns. First, when no corruption information is demonstrated at all, the model shows consistent evidence encoding with high confidence for the correct rule and low confidence for corrupted rules across layers. On the other hand, as corrupted demonstrations are introduced, probe confidence for corrupted rules rises significantly above baseline, indicating that models actively identify and encode the corrupting rule in the process.

Notably, internal representations of both correct and incorrect rules emerge most prominently in early-to-middle layers, with confidence levels becoming more stable in later layers. This suggests that models encode multiple competing rules simultaneously during the intermediate processing stages,

potentially creating an internal representational conflict that needs to be resolved to produce coherent final predictions.

## 4.2 LLMs Resolve Conflicting Evidence in Late Layers

While probes reveal where models encode information, they cannot determine when models actively use the information for predictions (Elazar et al., 2021). To observe the dynamics of model's favoring specific rules during ICL inference, we employ logit lens by projecting internal representations through the unembedding matrix to decode model predictions at intermediate layers (nostalgebraist, 2020). In more detail, to make the unembedded predictions more interpretable, we modify our ICL task prompts to elicit LLMs to identify the underlying rule rather than the final result.

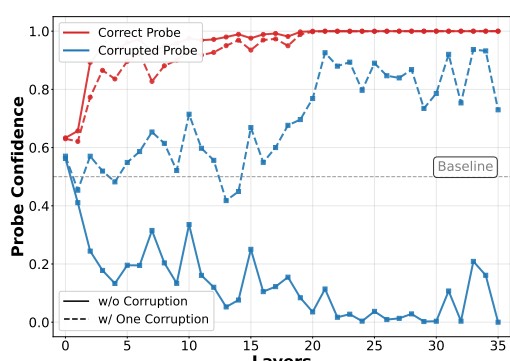

Figure 4: Linear probe confidence across model layers under different corruption scenarios. The **Correct Probe** detects the ground-truth operator while the **Corrupted Probe** detects incorrect operators under corruptions.

For consistency, we collect logit lens outputs under the same two corruption scenarios as linear probes. First, we notice a sharp transition in prediction probability occurring at late layers, where the model suddenly develops strong confidence for specific rules. In contrast, we observe that the model empirically exhibits minuscule prediction confidence across both correct and corrupted rules in early layers, and thus we omit those layers in Figure 5. Secondly, similar to probing results, we observe the simultaneous emergence of confidence for both correct and corrupted rules under the only-one-corruption setup, further indicating an existing competition between conflicting rules.

## 4.3 A Two-Stage Hypothesis of ICL Reasoning Dynamics under Corruption

Synthesizing the linear probe and logit lens findings, we observe a temporal separation between rule encoding (early-middle layers) and prediction formation (late layers). This temporal structure prompts a hypothesis that model's reasoning under in-context corruptions involves two complementary computational phases, which we term *conflict creation* and *conflict resolution*.

We characterize this two-stage hypothesis in more detail. In the first phase, certain model components create systematic weak points by encoding corrupted information, making the system vulnerable to conflicts. In the second phase, the conflict resolution process itself can fail when components are overly susceptible to minority corrupted evidence, leading to incorrect final predictions despite majority support for the correct rule. This framework predicts that early-layer heads should exhibit high sensitivity to positional corruption (creating vulnerabilities), while late-layer heads should demonstrate susceptibility patterns that undermine robust majority-based decisions (resolution failures). We test this hypothesis by identifying

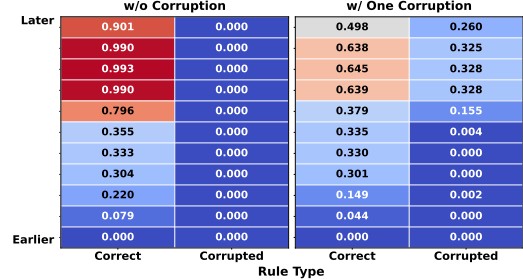

Figure 5: Logit lens predictions across corruption scenarios. Rows show different layers, columns represent correct and corrupted rules, and values indicate prediction probability for each rule decoded from layer-wise residual streams.

components associated with each failure mode and examining their causal contributions to reasoning under corruption.

### 4.3.1 Locating Conflict Creation Mechanisms

We begin by analyzing the first phase of our hypothesis: how certain model components create systematic vulnerabilities under corruption. Under ideal reasoning, each demonstration should

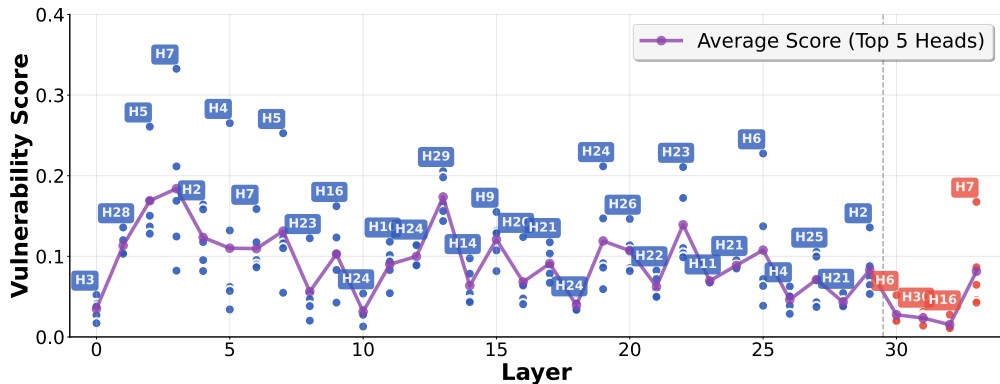

Figure 7: Distribution of Vulnerability Heads across model layers. The purple line shows the average vulnerability score (A × S) among the top 5 heads per layer. Individual heads are shown as dots.

contribute equally to rule inference due to the demonstration-dependent nature and equivalent complexity across demonstrations, yielding uniform positional effects.

However, we observe systematic positional variances in performance degradation as shown in Figure 2. This positional bias is not new to research, with prior studies attributing it mostly to the disproportionate integration from demonstrations through the attention mechanism (Han et al., 2024; Peysakhovich & Bastani, 2023; Chen et al., 2023). Building on prior work, we characterize two measurable attributes of attention heads that potentially contribute to the position-variant degradation under corruption: (1) disproportionate attention allocation per position, and (2) high output sensitivity when heavily-attended positions are corrupted. In other words, these heads are prone to create systematic vulnerabilities when contradictory information appears at their heavily-attended positions. To quantify them, we follow the ICL prompt segmentation from Cho et al. (2024), and focus our analysis on the final query forerunner token, which is identified critical for encoding input text representations during in-context learning.

**Measuring Positional Attention Allocation.**
For each attention head $(l, h)$ at layer $l$ and head $h$, we extract the attention score $\mathbf{A}_{l,h} \in \mathbb{R}^{S \times S}$ where $S$ represents sequence length. Following the demonstration-query segmentation (Cho et al., 2024), we compute the attention allocation from the query forerunner token position $\mathbf{F}$ to each demonstration position $p \in \{1, 2, \ldots, k\}$:

$$A_{l,h}^{(p)} = \frac{1}{|D_p|} \sum_{t \in D_p} \mathbf{A}_{l,h}[\mathbf{F}, t] \qquad (2)$$

where $D_p$ denotes the set of token positions belonging to demonstration $p$, and $|D_p|$ represents the number of tokens in that demonstration. This averaging yields per-demonstration attention scores that capture how much each head attends to specific positions during query processing.

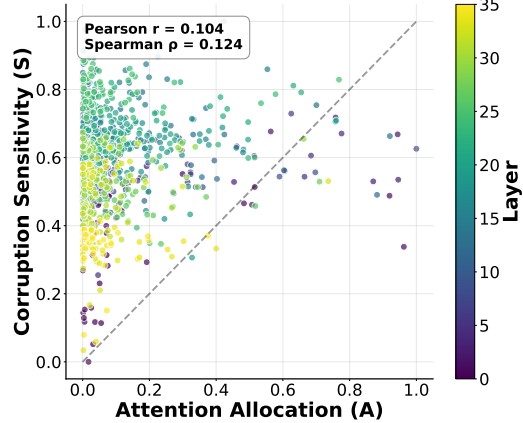

Figure 6: Weak correlation revealed between allocation (A) and corruption sensitivity (S).

**Measuring Attention Head Sensitivity to Corruption.** For each attention head $(l, h)$ and demonstration position $p$, we compute sensitivity as the normalized change in head outputs when that position is corrupted. Specifically, we first extract the head's output $\mathbf{o}_{l,h}^{(\mathbf{F})}$(clean) at the query forerunner position $\mathbf{F}$ under all-correct conditions, then corrupt only position $p$ with incorrect rule variants $r \in \mathcal{R}_{\text{wrong}}$ while keeping all other positions unchanged. We measure the output deviation $\mathbf{o}_{l,h}^{(\mathbf{F})}$(corrupted) for each corruption and compute the average normalized change. The sensitivity score is formulated as:

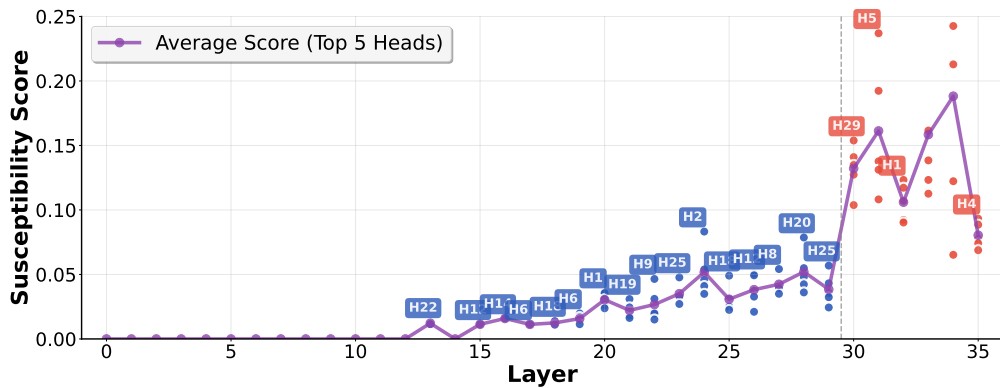

Figure 8: Distribution of susceptible heads across model layers. The purple line shows average susceptibility scores among top 5 heads per layer. Individual heads shown as dots.

$$S_{l,h}^{(p)} = \frac{\left\| \mathbf{o}_{l,h}^{(\mathbf{F})}(\text{corrupt}) - \mathbf{o}_{l,h}^{(\mathbf{F})}(\text{clean}) \right\|_2}{\left\| \mathbf{o}_{l,h}^{(\mathbf{F})}(\text{clean}) \right\|_2 + \epsilon} \tag{3}$$

High sensitivity indicates heads whose outputs fluctuate substantially with conflicting demonstrations. Intuitively, for corrupted position $p$, we are interested in localizing those attention heads that attend heavily to $p$ while being extremely sensitive in outputs to the single corruption at $p$. Before combining these attributes, we first assess whether they measure the same or distinct facets of attention head behaviour. To this end, we quantify their dependence using Pearson and Spearman correlations, with results shown in Figure 6. We observe weak associations across heads, indicating that our two characterizations are largely complementary rather than redundant. Thus, we multiply them as the *Positional Vulnerability Score* and serve it as a filter for salient heads.

Figure 7 shows the distribution of Positional-Vulnerability Heads, with blue indicating early-to-middle layer heads and red indicating late-layer ones. We observe a clear stratification with vulnerability behavior concentrated in early layers, supporting the first phase of our two-stage framework. However, we point out that this score only reflects correlational patterns rather than causal impacts.

To validate the causal role of these heads in creating vulnerabilities, we conduct targeted ablation experiments by masking identified vulnerability heads during inference. Ablations are conducted per position then averaged, leveraging the positional specificity of our metrics, and we run three rollouts for random corruption rules and random heads. As shown in Table 1, ablating top vulnerability heads improves model performance under corruption, with Qwen3-8B achieving up to 10.69% relative improvement when masking just 3 heads. Moreover, we notice that ablating vulnerability heads moderately reduces positional bias in corruption sensitivity, as shown in Table 3. This dual benefit helps validate both the specificity of our identification method and the systematic contributions of vulnerability heads to reasoning vulnerabilities under corruption.

### 4.3.2 LOCATING CONFLICT RESOLUTION MECHANISMS

We next investigate the second phase of our hypothesis: how certain model components fail at conflict resolution. Under our framework, robust conflict resolution requires maintaining support for correct predictions even when minority corrupted evidence appears, and we characterize the susceptibility to corruption as the opposite pattern of being easily swayed. Intuitively, and also established by previous studies, attention heads are the primary substrate for mediating how models selectively incorporate information from different demonstrations (Olsson et al., 2022; Halawi et al., 2023; Han et al., 2024). We therefore focus our analysis on identifying attention heads that exhibit this susceptibility pattern.

**Measuring Susceptibility via Logit Attribution.** We employ logit attribution (Wang et al., 2022) to quantify each attention head's contribution to final predictions under the same two corruption scenarios as above. For each query, we construct two contrastive contexts: clean context where all demonstrations are correct, and minority corruption context where only one demonstration is corrupted. We extract each attention head's output at the query forerunner position and compute its

Table 1: Attention head ablation results showing relative performance improvement (%) when masking different numbers and different types of attention heads.

| Models | Heads | # of Ablated Heads | | | |
|--------|-------|------|------|------|------|
| | | **2** | **3** | **5** | **10** |
| **Qwen3-4B** | Vulnerability | $+2.69_{\pm0.49}$ | $+2.69_{\pm0.48}$ | $+3.52_{\pm0.60}$ | $+0.35_{\pm0.32}$ |
| | Susceptible | $+1.24_{\pm0.38}$ | $+1.76_{\pm0.41}$ | $+1.03_{\pm0.29}$ | $+1.76_{\pm0.44}$ |
| | Random | $+0.83_{\pm0.33}$ | $-0.24_{\pm0.10}$ | $-2.07_{\pm0.85}$ | $-4.24_{\pm0.33}$ |
| **Qwen3-8B** | Vulnerability | $+6.24_{\pm0.51}$ | $+10.69_{\pm0.43}$ | $+6.48_{\pm0.37}$ | $+3.34_{\pm0.58}$ |
| | Susceptible | $+4.49_{\pm0.47}$ | $+5.52_{\pm0.53}$ | $+4.07_{\pm0.36}$ | $+3.52_{\pm0.61}$ |
| | Random | $+0.14_{\pm0.26}$ | $+0.58_{\pm0.35}$ | $-2.15_{\pm0.19}$ | $-5.43_{\pm0.42}$ |

attribution to the token of the correct rule, measured by prediction logits, through the unembedding matrix. Details of logit lens are discussed in Appendix D.

Then, we define each head's *Susceptibility Score* as the reduction in its contribution to the correct rule when exposed to minority corruption. A high positive score indicates that the head is easily swayed by conflicting evidence despite the overwhelming majority supporting the correct rule, suggesting problematic susceptibility rather than robust conflict resolution. We term heads with high susceptibility scores *Susceptible Heads*.

Figure 8 reveals a clear pattern that susceptible heads are predominantly concentrated in late layers, with the most susceptible heads achieving susceptible scores up to 0.25 (approximately 8% of total logit magnitude). This late-layer concentration contrasts sharply with early-layer vulnerability heads, supporting clear functional specialization in our two-phase framework. Since logit attribution does not guarantee causal influence, we validate these findings through targeted ablation experiments. As shown in Table 1, ablating susceptible heads also improves performance under corruption consistently, confirming their causal role in reasoning failures.

## 5 DISCUSSION

### 5.1 CROSS-RULE AND CROSS-TASK CONSISTENCY OF TARGETED HEADS

To evaluate whether our identified heads capture generalizable conflict-processing mechanisms, we measure the consistency of both Vulnerability and Susceptible Heads across different rules within Operator Induction tasks and across different task types. We compute Jaccard Similarity (Niwattanakul et al., 2013) for the top 20 heads identified under different conditions on Qwen3-8B. As shown in Table 4 at Appendix G, Vulnerability Heads exhibit stronger consistency than Susceptible Heads, with higher similarity across different rules (up to 0.818) than across different tasks. Specifically, attention heads L13H29 and L3H6 consistently rank among the top Vulnerability Heads across all evaluated scenarios, while L31H5 ranks among the top Susceptible Heads in five of six scenarios. We speculate that the lower consistency in Susceptible Heads reflects task-dependent information routing in late layers, whereas the conflict-encoding mechanism handled by Vulnerability Heads in early layers may be more universal across scenarios. This moderate cross-rule and cross-task consistency suggests that the identified heads capture partially generalizable mechanisms.

### 5.2 REASONING UNCERTAINTY UNDER DEMONSTRATION CONFLICTS

Beyond classification accuracy, we also examine whether models exhibit calibrated reasoning uncertainty when facing demonstration conflicts. Since our rule inference tasks employ Chain-of-Thought (CoT) reasoning, we track next-token prediction entropy during CoT generation as a proxy for the model's internal uncertainty, following recent work showing that entropy reliably reflects and monitors reasoning confidence (Fu et al., 2025; Kang et al., 2025; Zur et al., 2025).

We measure the next-token entropy for Qwen3-8B on the Operator Induction queries under clean, corrupted, and corrupted-with-intervention conditions. As shown in Table 5 at Appendix G, demonstration conflicts significantly increase reasoning uncertainty: entropy rises from 0.0952 to 0.1161 nats, indicating the model appropriately reduces confidence when facing conflicting information. Moreover, ablating identified heads partially restores confidence: masking the top-5 Vulnerability Heads and Susceptible Heads both reduce the next-token entropy, approaching the clean baseline.

This demonstrates that our identified components causally contribute not only to prediction accuracy but also to elevated reasoning uncertainty, providing additional validation of their mechanistic roles through an orthogonal evaluation metric.

### 5.3 EMPIRICAL SYNERGY BETWEEN THE TWO PHASES

We are also interested in the synergy among these components within the model's computational pathway. Specifically, we examine whether ablating vulnerability heads reduces the susceptibility of late-layer heads by measuring changes in their logit contributions, and compare these effects against random head ablations. As shown in Figure 12 at Appendix G, ablating vulnerability heads reduces the susceptibility scores of top susceptible heads (e.g. L30H29 and L31H5) significantly compared with random ablation. This case study suggests an empirical synergy between early-layer vulnerability creation and late-layer susceptibility, through the mediation of the two types of heads.

### 5.4 VULNERABILITY HEADS CONTRIBUTE TO POSITIONAL BIAS

Beyond performance improvement, we also notice that ablating vulnerability heads reduces positional bias in corruption sensitivity. We formally define positional bias as $\text{PB} = \text{Var}_{p=1}^{k}[\Delta\text{Acc}_p]$, where $\Delta\text{Acc}_p$ represents the accuracy decrease when position $p$ is corrupted. As shown in Table 3 at Appendix G, removing vulnerability heads reduces positional variance moderately across models. Since vulnerability heads exhibit disproportionate attention to certain positions while showing high sensitivity to corruption at those positions, their removal likely mitigates the uneven impact of corruption across demonstration sequences, leading to more uniform vulnerability patterns.

### 5.5 LIMITATIONS

First, our controlled experimental framework requires principled constraints to isolate and analyze conflict resolution mechanisms. However, real-world rule inference tasks can exhibit more complexity, with interdependent demonstrations and varying evidence weights. Thus, while our ablation studies exhibit mitigation for performance degradation, we view them primarily as mechanistic validation of the causal role of our identified model components, rather than claiming as practical solutions.

Second, understanding exact causal relationships between cross-layer attention heads remains challenging and requires sophisticated circuit analysis techniques beyond our current methodology (Sharkey et al., 2025; Ameisen et al., 2025). While our analyses provide initial evidence for component interactions, we take the deeper mechanistic characterization of how truths and conflicts cascade through the computational processing as important future work (e.g. cross-layer transcoders (Dunefsky et al., 2024) for rule circuits identification) .

## 6 CONCLUSION

We investigate how LLMs process conflicting demonstrations during in-context rule inference. Using tasks that require genuine demonstration reliance and demonstration modularity, we establish a controlled framework for studying inter-context conflicts through position-specific corruptions. Our experiments reveal that models suffer substantial performance degradation from single corrupted demonstrations, with the corrupted rule systematically accounting for 80% of wrong predictions despite appearing in only one position among correct ones.

Through linear probes and logit lens analysis, we uncover a two-phase computational structure: models encode both correct and corrupted rules in early-to-middle layers but develop prediction confidence only in late layers, suggesting temporal separation between conflict creation and resolution. Then, we identify specific attention heads responsible for each phase: Vulnerability Heads in early-to-middle layers determine where conflicts enter the system through positional attention bias and high corruption sensitivity, while Susceptible Heads in late layers determine whether models surrender to conflicts by reducing support for correct predictions when exposed to minority corrupted evidence. Finally, targeted ablation of these components improves performance under corruption by over 10%, validating the causal roles of the identified components. Our work establishes initial mechanistic foundations for understanding reasoning failures in demonstration-dependent rule inference tasks, providing a basis for future research on studying in-context learning conflict resolution.

## ETHICS STATEMENT

This research focuses on mechanistic interpretability of language models during in-context rule learning, aiming to understand how models process conflicting demonstrations. Our work employs established interpretability techniques on existing datasets and models to identify failure modes in reasoning under corruption. We do not anticipate harmful applications of this research, as it focuses on understanding and potentially mitigating model vulnerabilities rather than exploiting them. All experiments were conducted on publicly available models and datasets following standard research practices. The authors have read and adhere to the ICLR Code of Ethics.

## REPRODUCIBILITY STATEMENT

To support reproducibility, we provide our codebase at `https://anonymous.4open.science/r/Understanding-Rule-Inference-23EE`, including implementations for corruption-based interventions, linear probe training, logit lens analysis, attention head identification, and ablation experiments. Detailed experimental configurations and evaluation protocols are documented in the appendix. Our experiments use publicly available datasets (Operator Induction tasks) and models (Qwen3 and Llama-3.1 series) with standard computational requirements.

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

## A LLMS USAGE IN THE PAPER

LLMs were used only occasionally to help polish the writing (propose new words, grammar and spelling correction). All technical ideas, experimental designs, analyses, conclusions, writing were developed and carried out entirely by the authors. The authors have full responsibility for the final text.

## B DETAILS OF CORRUPTION-BASED INTERVENTION FRAMEWORK

### B.1 CORRUPTION-BASED EXPERIMENTAL DESIGN

Our evaluation protocol systematically examines model robustness to conflicting demonstrations through controlled corruption at specific positions within the demonstration sequence. We implement a paired comparison design where each query receives identical corrupted demonstrations under two distinct instructional conditions: *warned* (explicit guidance about potential errors) versus *unwarned* (standard ICL prompts).

### B.1.1 DATA SCALE AND COVERAGE

For each experimental configuration, we evaluate:

**Operator Induction Tasks.**

- **Query samples**: 60 unique queries per task
- **Shots per query**: 4 demonstrations
- **Corruption positions**: 4 positions
- **Rollouts per position**: 3 independent trials with different random seeds
- **Total evaluations**: 1,440 paired comparisons ($60 \times 4 \times 3 \times 2$ conditions)

**Fake Word Inference Task.**

- **Query samples**: 100 unique target pairs (color-object combinations)
- **Shots per query**: 4 demonstrations
- **Corruption positions**: 4 positions
- **Rollouts per position**: 3 independent trials with different random seeds
- **Total evaluations**: 2,400 paired comparisons ($200 \times 4 \times 3$)

Note that in our tasks the choice of corrupted rules can be flexible based on the hypothesis rule space. This design yields robust statistical estimates across position-specific corruption patterns while controlling for demonstration-level confounds through identical corruption application across warned and unwarned conditions. Table 2 validates our task selection criteria for studying demonstration-dependent conflict resolution. All tasks exhibit near-chance 0-shot performance, confirming genuine demonstration reliance. Performance substantially improves with few-shot demonstration, demonstrating reliable ICL capability. This pattern of models' failing without demonstrations yet succeeding with them intrinsically satisfies our principled framework requirement that models must genuinely rely on contextual evidence rather than parametric knowledge.

### B.1.2 CORRUPTION MECHANISM

We implement systematic single-position corruption where exactly one demonstration at position $i$ (e.g., $i \in \{0, 1, 2, 3\}$ for 4-shot scenarios) receives a corrupted rule while maintaining the majority rule principle. For Operator Induction tasks, corruption involves replacing the correct mathematical operator with an alternative operator (+, -, ×), ensuring the corrupted demonstration exhibits a different underlying rule while maintaining surface-level plausibility. For Fake Word Inference, corruption involves substituting the correct fake word mapping with an incorrect alternative from the same category (e.g., replacing the correct color mapping with a different color's fake word).

Table 2: Clean (Uncorrupted) ICL baseline performance across tasks and models.

| Task | Model | 0-shot | 4-shot | 6-shot | 8-shot |
|------|-------|--------|--------|--------|--------|
| **Operator Induction** | Qwen3-4B | 0.333 | 0.850 | 0.750 | 0.783 |
| | Qwen3-8B | 0.233 | 0.767 | 0.800 | 0.767 |
| | Qwen3-14B | 0.283 | 0.867 | 0.800 | 0.850 |
| | Qwen3-32B | 0.233 | 0.933 | 0.983 | 0.933 |
| | Llama-3.1-8B | 0.333 | 0.733 | 0.667 | 0.667 |
| **Operator Induction (Interleaved)** | Qwen3-4B | 0.350 | 0.833 | 0.767 | 0.800 |
| | Qwen3-8B | 0.250 | 0.783 | 0.817 | 0.783 |
| | Qwen3-14B | 0.300 | 0.883 | 0.817 | 0.867 |
| | Qwen3-32B | 0.250 | 0.950 | 1.000 | 0.950 |
| | Llama-3.1-8B | 0.350 | 0.750 | 0.683 | 0.683 |
| **Fake Word Inference** | Qwen3-4B | 0.000 | 0.295 | 0.335 | 0.375 |
| | Qwen3-8B | 0.025 | 1.000 | 1.000 | 1.000 |
| | Qwen3-14B | 0.010 | 1.000 | 1.000 | 1.000 |
| | Qwen3-32B | 0.070 | 0.985 | 1.000 | 0.990 |
| | Llama-3.1-8B | 0.000 | 0.165 | 0.195 | 0.110 |

## B.2 MODEL RESPONSE EVALUATION PROTOCOL

### B.2.1 PROMPT ENGINEERING

**Operator Induction: Baseline (Unwarned) Condition.** Taking 4-shot in-context rule inference as an example, the baseline condition employs standard formatting from (Zong et al., 2024) without explicit guidance about potential demonstration conflicts, where we introduce a targeted corruption at position 2:

```
The text contains two digit numbers and a ? representing the
mathematical operator. Induce the mathematical operator (addition,
multiplication, minus) according to the results of the in-context
examples and calculate the result. Reason carefully step by step
and provide the final answer.

Support Set:
8 ? 6 =
Answer: 14

3 ? 5 =
Answer: 8

7 ? 2 =
Answer: 5

9 ? 4 =
Answer: 13

Question:
6 ? 3 = ?

Answer:
```

**Operator Induction: Upper Bound (Warned) Condition.** The warned condition provides explicit instructions for conflict resolution through majority reasoning:

```
The text contains two digit numbers and a ? representing the
mathematical operator. Induce the mathematical operator (addition,
multiplication, minus) according to the results of the in-context
examples and calculate the result. IMPORTANT: Follow these steps:
1) Look at each example and identify what operation it shows
2) Count how many examples show addition (+), subtraction (-),
   and multiplication (*)
3) The operation that appears most frequently is the correct one
4) Apply that operation to solve the final problem
Reason carefully step by step and provide the final answer.

Support Set:
[Same as baseline condition]

Question:
6 ? 3 = ?

Answer:
```

**Fake Word Inference: Baseline (Unwarned) Condition.** For the Fake Word Inference task, the baseline condition presents demonstrations mapping fake words to real colors and objects without explicit conflict resolution guidance:

```
You will see examples showing what fake words mean in terms of
real colors and objects. Learn the mapping from fake words to
real words, then answer the question using real words only.
Think step by step, then provide your final answer in the format:
<answer>your answer</answer>

Examples:
blimontar vundelka -> red sheep
xelvorian glemorax -> blue hat
blimontar plintovar -> red scarf
pflumenic glemorax -> green hat

Question:
blimontar glemorax -> ?

Answer:
```

**Fake Word Inference: Upper Bound (Warned) Condition.** The warned condition includes explicit instructions for identifying the majority mapping:

```
You will see examples showing what fake words mean in terms of
real colors and objects. Learn the mapping from fake words to
real words, then answer the question using real words only.
IMPORTANT: Follow these steps:
1) Look at each example and identify what each fake word means
2) Count how many examples show each meaning
3) The meaning that appears most frequently is the correct one
4) Apply that meaning to answer the question with real words
Think step by step, then provide your final answer in the format:
<answer>your answer</answer>

Examples:
[Same as baseline condition]

Question:
```

```
blimontar glemorax -> ?

Answer:
```

### B.2.2 EVALUATION

We employ vLLM framework (Kwon et al., 2023) for efficient batched inference on two NVIDIA A100 80 GB GPUs. We implement the primary evaluation approach using Qwen2.5-7B-Instruct for LLM-as-the-judge (Li et al., 2024).

**Operator Induction Evaluation.** We employ the following standardized prompt template:

```
You are evaluating a mathematical reasoning response.

The task is: Given texts showing mathematical expressions with
unknown operators, induce the operator and calculate the result.

Expected Answer: {expected_answer}

Model Response: "{response}"

Question: Does the model's response reflect the correct final
answer of {expected_answer}?

Look for the final numerical answer in the response, even if the
reasoning process contains errors or is verbose. The response is
correct if it arrives at the final answer {expected_answer},
regardless of the path taken.

Answer with exactly one word: "YES" if the response contains the
correct answer {expected_answer}, "NO" if it does not.
```

**Fake Word Inference Evaluation.** For Fake Word Inference, we extract answers from the structured `<answer>...</answer>` format and perform exact string matching against ground truth color-object pairs (e.g., "red hat"). This deterministic evaluation ensures consistent assessment of mapping accuracy.

During performance degradation evaluation under our corruption framework, we conform to the following two characteristics for statistic reports:

**Majority Rule Verification.** Our corruption mechanism maintains the majority rule principle: in 4-shot scenarios with single-position corruption, correct demonstrations maintain at least a 3:1 majority over corrupted ones, providing a theoretically sound basis for expecting successful conflict resolution under ideal conditions. This applies to both Operator Induction tasks (correct operator appears in 3 of 4 demonstrations) and Fake Word Inference (correct mapping appears in 3 of 4 examples). This comprehensive protocol enables systematic measurement of model vulnerability to conflicting evidence while controlling for instruction-dependent performance variations, yielding interpretable estimates of real-world autonomous conflict detection and resolution scenarios.

**Paired Experimental Control.** Critical to our methodology is the paired nature of evaluations: for each (query, position, rollout) tuple, both warned and unwarned conditions receive identical demonstrations with identical corruption patterns. Specifically, we use position-specific random seeds to ensure identical corruptions across warned/unwarned conditions for valid paired comparisons. This design isolates the effect of explicit conflict resolution instructions from confounds related to demonstration content or corruption severity.

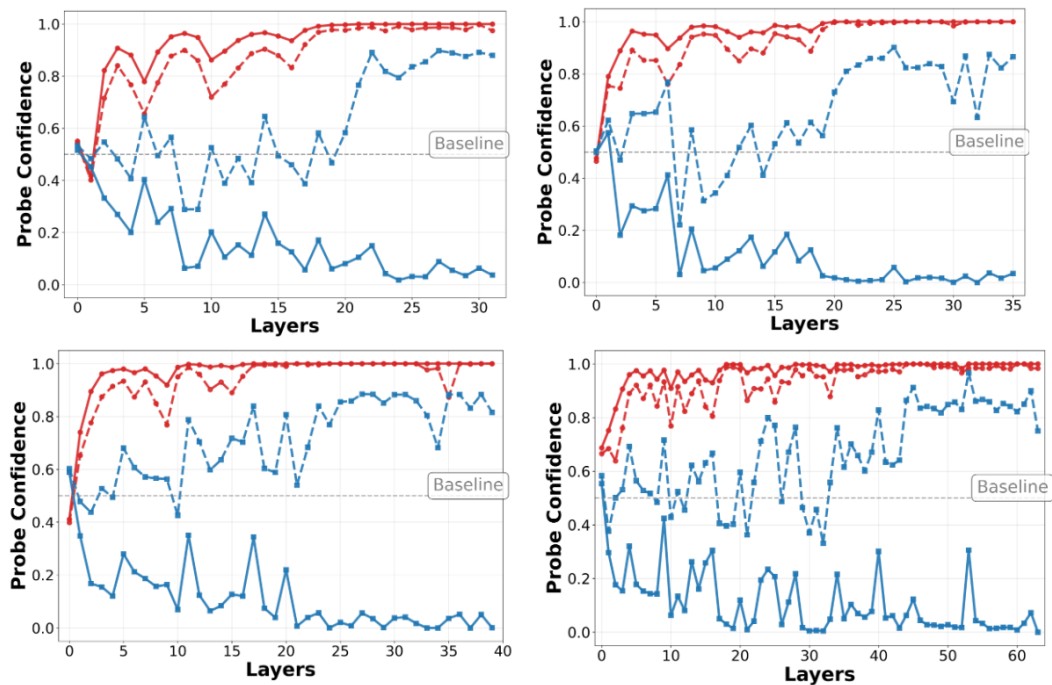

Figure 9: Linear probe confidence across model layers for Qwen3-4B, Qwen3-8B, Qwen3-14B, and Qwen3-32B from left to right, up to down. **Red solid**: Correct-rule probe confidence in all-correct scenarios. **Red dashed**: Correct-rule probe confidence in one-corrupted scenarios. **Blue solid**: Corrupted-rule probe confidence in all-correct scenarios. **Blue dashed**: Corrupted-rule probe confidence in one-corrupted scenarios. The baseline represents chance-level performance.

## C  LINEAR PROBE ANALYSIS DETAILS

### C.1  PROBE TRAINING METHODOLOGY

We train linear probes as binary classifiers to detect the presence of specific mathematical operators (+, -, ×) in model internal representations. Each probe learns to answer "Is the target rule present in the demonstrations?" rather than performing multi-class operator identification. This design provides a principled approach to measure rule encoding strength across different corruption scenarios.

For each target operator, we generate 600 balanced training samples, containing 300 positive samples for the target rule. Take the Operator Induction task under 4-shot ICL as an example, we compose the positive samples 75 each with 1/4, 2/4, 3/4, 4/4 target demonstrations, and target rules are completely absent in the 300 negative samples. Then, demonstrations are randomly shuffled to eliminate positional effects, ensuring probes learn rule content rather than position patterns. We train simple layer-wise linear probes with L1-regularization (Alain & Bengio, 2016) on the residual streams of the LLMs. Specifically, for each layer $l$, we predict:

$$\hat{y}^{(i)} = \sigma(\mathbf{w}^T \mathbf{h}_l^{(i)} + b) \tag{4}$$

where $\mathbf{h}_l^{(i)}$ is the representation at layer $l$ for sample $i$, and we optimize over the regularized objective.

### C.2  TWO-SCENARIO EVALUATION PROTOCOL

We evaluate trained probes under two controlled scenarios to measure rule encoding dynamics: all Correct and only one corrupted. In the first scenario, all 4 demonstrations exhibit the same target rule (e.g., addition); in the second scenario, 3 demonstrations show the target rule, while 1 demonstration shows conflicting rule (e.g., multiplication or minus)

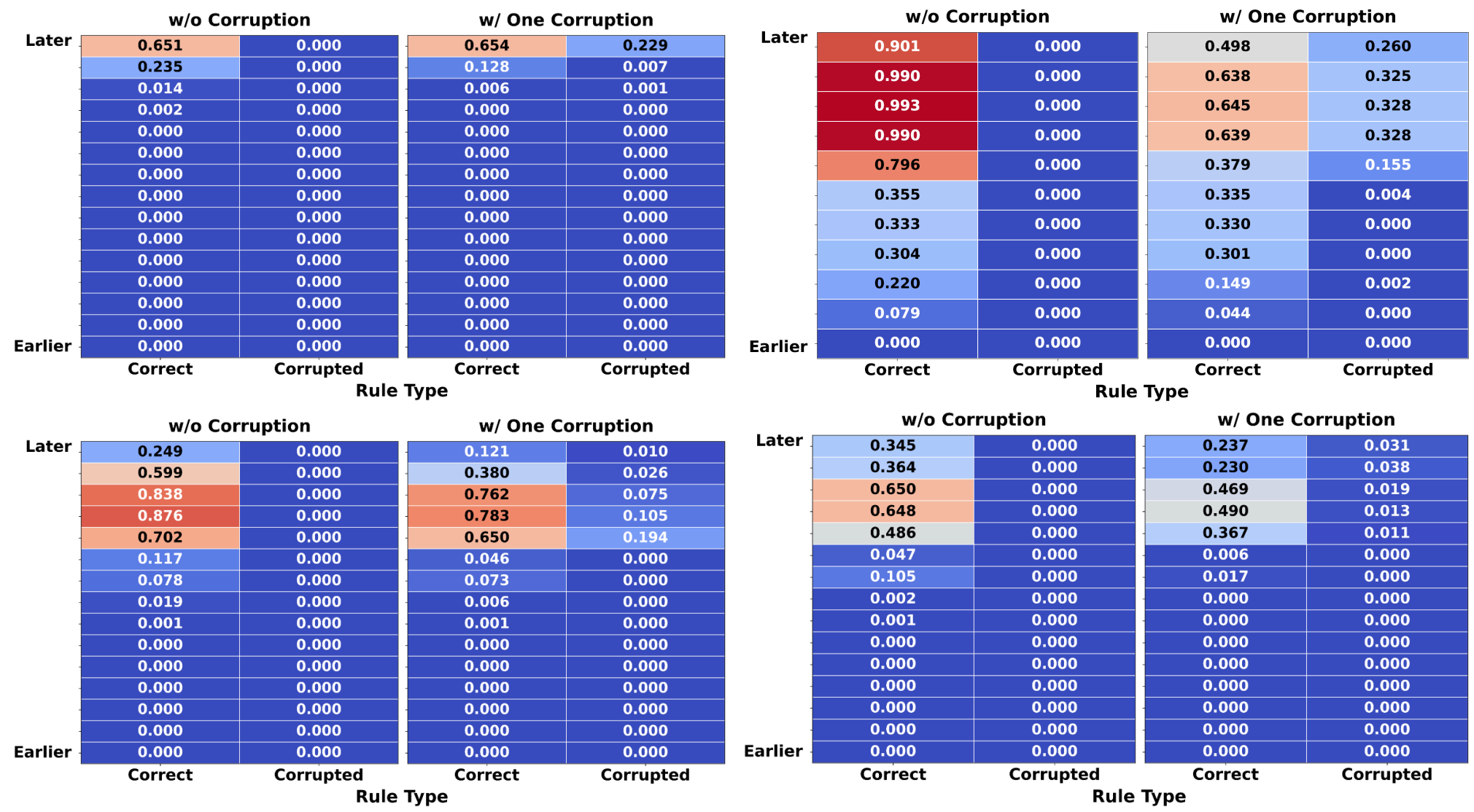

Figure 10: Logit lens results across model layers for Qwen3-4B, Qwen3-8B, Qwen3-14B, and Qwen3-32B from left to right, up to down.

For each scenario, we extract representations from all layers and compute probe confidence as the measurement for model's encoded confidence of each rule. This yields four evaluation curves: Correct-rule probe confidence in all-correct scenario, correct-rule probe confidence in one-corrupted scenario, corrupted-rule probe confidence in all-correct scenario, and corrupted-rule probe confidence in one-corrupted scenario. To encourage the model to generate the encoded rule instead of the target answer directly, we modify the prompt to elicit direct operator prediction rather than numerical calculation. Specifically, for the Operator Induction tasks, we replace the query format from "6 ? 3 = ?" to:

```
What mathematical operation does ? represent?
Choose from: plus, minus, multiplication

Answer:
```

Figure 9 illustrates these probe confidence patterns across three different model sizes. The results consistently demonstrate our key findings across model architectures. The probe analysis reveals two key mechanistic insights. First, in one-corrupted scenarios, both correct and corrupted rule probes exhibit above-baseline confidence, demonstrating that models encode multiple competing rules simultaneously rather than selecting a single rule early in processing. Second, rule encoding emerges prominently in early-to-middle layers (layers 5-20), with confidence levels stabilizing in later layers.

## D Logit Lens Analysis Details

We employ logit lens analysis (nostalgebraist, 2020) to track how models progressively resolve conflicting evidence during in-context learning. The logit lens projects intermediate representations from each transformer layer through the final layer normalization and language modeling head to observe predicted token probabilities at different processing stages.

For a hidden state $\mathbf{h}_\ell$ at layer $\ell$, the logit lens computes:

$$\mathbf{p}_\ell = \text{softmax}(\mathbf{W} \cdot \text{LayerNorm}(\mathbf{h}_\ell))$$

where $\mathbf{W}$ is the unembedding matrix of the language modeling head and $\mathbf{p}_\ell$ represents the probability distribution over vocabulary tokens as predicted from layer $\ell$.

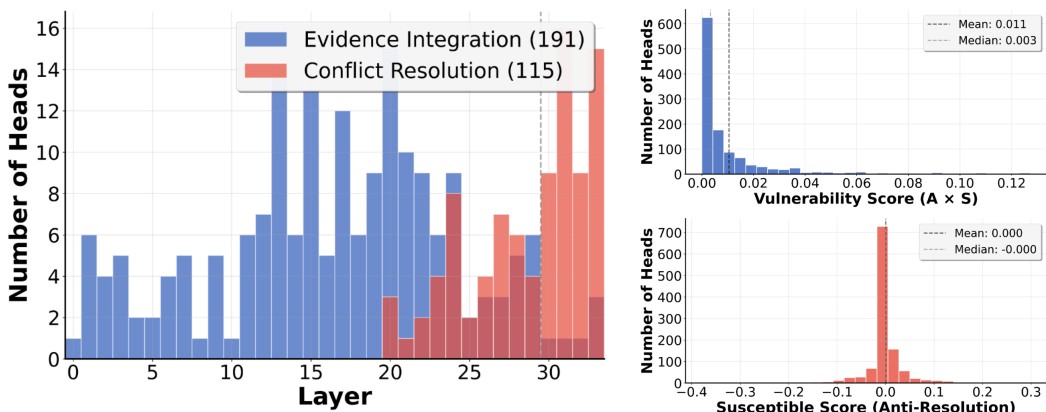

Figure 11: Distribution statistics of vulnerability and susceptible heads.

Then, we focus specifically on mathematical operator prediction by analyzing the probability assigned to operator words at each layer. Our analysis tracks two primary quantities as below:

**Correct Rule Probability** The probability assigned to the ground-truth mathematical operator word at layer $\ell$:

$$P_\ell(\text{correct}) = \max_{t \in T_{\text{correct}}} \mathbf{p}_\ell[t]$$

where $T_{\text{correct}}$ includes all tokens that match or prefix the correct operator word (Wendler et al., 2024).

**Corrupted Rule Probability** The probability assigned to the operator introduced through minority corruption:

$$P_\ell(\text{corrupted}) = \max_{t \in T_{\text{corrupted}}} \mathbf{p}_\ell[t]$$

We evaluate models under controlled corruption scenarios, exactly the same to the probes analysis, to measure conflict resolution dynamics internally. Wer show different models' logit lens decoding results in Figure 10. The logit lens analysis provides direct evidence for the second leg of our central hypothesis: models show strong signal of resolving conflicts in late layers.

## E    THE STATISTICS OF TWO TYPES OF HEADS

The layer-wise distribution in Figure 11 confirms the temporal separation predicted by our two-phase hypothesis, with vulnerability heads predominantly appearing in layers 0-20 and susceptible heads concentrated in layers 25-35. More importantly, the score distributions reveal that our metrics are highly selective rather than universal - the vast majority of attention heads exhibit near-zero scores (median values approach zero for both metrics), while only a small subset shows high vulnerability or susceptibility. This concentration of high scores among a minority of heads validates that our identification methodology successfully isolates functionally specialized components rather than capturing generic attention behaviors. Combined with our causal ablation results, these statistics provide strong evidence that the identified heads play specific, non-redundant roles in creating and failing to resolve conflicts during rule inference.

## F    WHY LOGITS ATTRIBUTION OVER ACTIVATION PATCHING FOR LOCATING CONFLICT RESOLUTION HEADS

Activation patching is another widely-used technique to identify causally important model components by targetedly replacing their activations and observing performance changes (Meng et al., 2022; Zhang & Nanda, 2023). While our experimental setup naturally supports activation patching, where we have structured clean and corrupted demonstrations, we employ logits attribution to localize

Table 4: Jaccard Similarity for top 20 Vulnerability Heads (VH) and Susceptible Heads (SH) across rules and tasks on Qwen3-8B. Higher values indicate stronger consistency. OI stands for the Operator Induction task and FW stands for the Fake Word Inference task.

| VH | Within-Rule Consistency | | | | Cross-Task Consistency | | |
|---|---|---|---|---|---|---|---|
| | OI+ | OI− | OI× | OI | OI | OI-int | FW |
| OI+ | 1.000 | – | – | – | – | – | – |
| OI− | 0.538 | 1.000 | – | – | – | – | – |
| OI× | 0.667 | 0.667 | 1.000 | – | – | – | – |
| OI | 0.739 | 0.667 | **0.818** | 1.000 | 1.000 | – | – |
| OI-int | – | – | – | – | 0.538 | 1.000 | – |
| FW | – | – | – | – | 0.290 | 0.290 | 1.000 |
| **SH** | OI+ | OI− | OI× | OI | OI | OI-int | FW |
| OI+ | 1.000 | – | – | – | – | – | – |
| OI− | 0.333 | 1.000 | – | – | – | – | – |
| OI× | 0.290 | 0.290 | 1.000 | – | – | – | – |
| OI | 0.333 | 0.250 | 0.250 | 1.000 | 1.000 | – | – |
| OI-int | – | – | – | – | 0.290 | 1.000 | – |
| FW | – | – | – | – | 0.176 | 0.143 | 1.000 |

Table 5: Average next-token prediction entropy during rule inference tasks under different conditions on Qwen3-8B. Lower entropy indicates higher model reasoning confidence.

| Condition | Mean Entropy | Std |
|---|---|---|
| Clean | 0.0952 | 0.0239 |
| Corrupted | 0.1161 | 0.0217 |
| Corrupted + Vulnerability Heads masked | 0.1148 | 0.0167 |
| Corrupted + Susceptible Heads masked | 0.1120 | 0.0205 |

Resolution Heads for mechanistic reasons. Importantly, activation patching fails to isolate conflict resolution mechanisms due to the temporal separation of conflict detection and conflict resolution functions. For example, patching heads that encode conflict with clean activations simply relieves the necessity of conflict resolution, consequently creating systematic false positives. Empirically, we find moderate overlap between vulnerability heads and top activation patching heads (Jaccard similarity of 0.301 for the top-10), suggesting the conflation of two phases from activation patching. On the other hand, logits attribution respects this functional separation by measuring each head's direct contribution given the evidence it actually processes.

# G  ADDITIONAL RESULTS

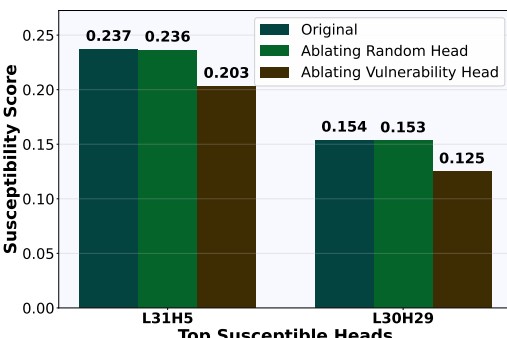

Figure 12: Ablating vulnerability heads reduces the susceptibility score of susceptible heads

Table 3: Relative positional variance reduction (%) when ablating vulnerability heads.

| Models | # of Ablated Heads | | |
|--------|------|------|------|
|        | 2    | 3    | 5    |
| Qwen3-4B | 18.45 | 22.50 | 19.73 |
| Qwen3-8B | 25.67 | 31.60 | 22.24 |

