# OpenReview forum: "Mechanistic Analysis of Demonstration Conflicts in In-Context Learning"
_ICLR.cc/2026/Conference — Submitted to ICLR 2026_

### Official Review · Reviewer_jDr5 · 2025-10-21

**Soundness:** 2
**Presentation:** 2
**Contribution:** 2
**Rating:** 4
**Confidence:** 3

**Summary:**

The paper explores how Large Language Models (LLMs) handle conflicting demonstrations in In-Context Learning (ICL), focusing on rule inference tasks (where models need to infer underlying patterns from examples). It is found that a single "corrupted demonstration" leads to a significant performance drop – although the corrupted rule appears in only one position, it accounts for 80% of incorrect predictions. Through linear probes and logit lens analysis, the study reveals a two-phase computational structure: LLMs encode both correct and incorrect rules in early network layers, but form prediction confidence only in late layers. In addition, two types of attention heads that cause reasoning failures are identified: "Vulnerability Heads" (in early-to-middle layers, exhibiting positional attention bias and high sensitivity to "corrupted (examples)"); and "Susceptible Heads" (in late layers, reducing support for "correct predictions" when exposed to corrupted evidence). "Targeted ablation" of these attention heads can improve performance by more than 10%, establishing a mechanistic understanding of how LLMs handle conflicting evidence in "contextual rule learning".

**Strengths:**

1. Mechanistic clarity: The study reveals the internal two-phase process of "early rule encoding + late confidence formation" and locates specific attention heads that cause failures, providing a practical mechanistic explanation for the "vulnerability of ICL to conflicts".
2. Rigorous validation: The results are consistent across multiple LLMs (Qwen3 series, Llama-3.1), task variants (with/without interleaved "operator induction" tasks), and analytical tools (linear probes, logit lens, targeted ablation), enhancing the reliability of the conclusions.
3. Practical relevance: The identified attention heads and "ablation strategy" provide specific improvement paths for enhancing the robustness of LLMs to "noisy demonstrations" in real ICL scenarios.

**Weaknesses:**

1. Task limitations: Focusing on "operator induction" (a type of mathematical rule task) limits the exploration of how conflicts affect other ICL tasks (such as text classification and commonsense reasoning).
2. Incomplete analysis of cross-layer causal pathways. Although the empirical synergistic effect between the two types of heads is found (shielding vulnerability heads can reduce the impact of susceptible heads), the complete computational circuit of information transmission and interaction across different layers has not been accurately characterized.
3. Model diversity gap: Although multiple open-source LLMs are tested, the generalization to "closed-source models (such as GPT-4)" or "ultra-large-scale LLMs" is not verified.
4. Mitigation strategies not explored: The paper proves that "ablating problematic attention heads" is effective, but does not propose proactive methods (such as training tricks and prompting strategies) to mitigate "demonstration conflicts" in practical scenarios.
5. Insufficient fine-grained analysis: How "model scale, pre-training objectives, or data filtering" affect "conflict handling" has not been deeply analyzed.
6. Single evaluation metric. The study mainly focuses on classification accuracy and does not explore other important dimensions such as the model's confidence calibration and uncertainty estimation when facing conflicts.

**Questions:**

See Weaknesses.

---

> ### Author Response · Authors · 2025-11-20
>
> Thank you for your careful review of our work. We really appreciate the points for potential improvement that you highlighted.
>
> **W1: Task Limitations**
>
> Firstly, we focus on rule-inference tasks for two reasons: 1) in these tasks LLMs have to rely on demonstrations to reach reliable performance and they can only achieve chance-level performance (by random guessing) without demos; 2) each demonstration is equally important. Thus, this setup blocks the model's pathways of relying solely on parametric knowledge—forcing the reliance on demonstrations to yield good ICL performance—and we can simply introduce conflicts by corrupting rules at specific positions without undermining others. Other popular ICL tasks, for example text classification and sequence completion, though available, are not ideal for our research and interpretation because 1) LLMs can reach far beyond chance-level in 0-shot for most text classification tasks; 2) Modifying elements at different positions to different amounts can have different degrees of impact in sequence completion tasks. These constraints are fundamental to our corruption framework and subsequent mechanistic analysis.
>
> Secondly, regarding task complexity: the goal of mechanistic interpretability is to isolate and understand specific computational phenomena, not to benchmark performance on hard problems. Simple and intuitive tasks [1, 2, 3] can often be preferable because they reduce confounds and make internal mechanisms more interpretable.
>
> That said, we did extend our analysis beyond operator induction. We added a fake word inference task that maps nonsense words to real concepts (e.g., "blimontar glemorax" → "red hat"). This task prevents memorization - models cannot have seen "blimontar" during pretraining and must learn the mapping from demonstrations. Importantly, it satisfies the same two constraints: models fail without examples and each demonstration is modular. We observed similar vulnerability patterns, as shown in the table below, and similar two-staged conflict processing in this task.
>
> |Model|4-shot|6-shot|8-shot|
> |---|---|---|---|
> |Qwen3-4B|15.45%|16.81%|10.62%|
> |Qwen3-8B|17.10%|14.06%|13.27%|
>
> To summarize, the essential criteria in our intervention framework are the demonstration reliance and demonstration modularity of the task, which formulates the inter-context ICL conflict in a principled way, and we take the rule inference tasks as a tractable start to observe the dynamics and provide initial mechanistic understandings. We will include a more detailed discussion and relevant results on the fake word inference task in our revised manuscript.
>
> **W2: Incomplete Analysis of Cross-layer Causal Pathways**
>
> Although demystifying the full causal information flow is intriguing and significant, current attempts in mechanistic interpretability are often regarded as “not faithful enough” and “can capture spurious correlations” on off-the-shelf LLMs [4]. Given these methodological constraints, we position our contributions as: establishing a foundational framework for studying demonstration conflicts in ICL and providing initial evidence of the two-phase structure. And we believe this represents a meaningful step forward while acknowledging future work for complete component-level understandings.
>
> Also, we have carefully addressed the discussion about the causal flow in the second paragraph of our Limitation section. Specifically, around line 475, we state that “While our analyses provide initial evidence for component interactions, we take the deeper mechanistic characterization of how truths and conflicts cascade through the computational processing as important future work”.
>
> [1] Levy A A, Geva M. Language models encode numbers using digit representations in base 10[C]//Proceedings of the 2025 Conference of the Nations of the Americas Chapter of the Association for Computational Linguistics: Human Language Technologies (Volume 2: Short Papers). 2025: 385-395.
>
> [2] Gurnee W, Tegmark M. Language Models Represent Space and Time[C]//The Twelfth International Conference on Learning Representations.
>
> [3] Engels J, Michaud E J, Liao I, et al. Not All Language Model Features Are One-Dimensionally Linear[C]//The Thirteenth International Conference on Learning Representations.
>
> [4] Sharkey L, Chughtai B, Batson J, et al. Open problems in mechanistic interpretability[J]. arXiv preprint arXiv:2501.16496, 2025.

---

> ### Author Response · Authors · 2025-11-20
>
> **W3: Generalization to "closed-source models" or "ultra-large-scale LLMs"**
>
> We appreciate this suggestion but want to clarify the fundamentals of mechanistic interpretability research. Closed-source models like GPT-4 are fundamentally incompatible with mechanistic analysis because they do not provide access to internal representations (layer-wise activations, attention patterns, residual streams) required for techniques like linear probing, logit lens, and component-level attribution. Without these, mechanistic interpretability cannot be performed. This is not a limitation of our study but an inherent constraint of the closed-source paradigm.
>
> Regarding ultra-large-scale models, mechanistic interpretability research typically prioritizes tractable model sizes where comprehensive layer-wise and component-wise analysis remains computationally feasible. While testing on 100B+ parameter models would be interesting, it introduces significant practical barriers (computational cost, harder to interpret massive internal spaces) without fundamentally changing the research question.
>
> **W4, W5: Proactive Mitigations and “Fine-grained” Analysis**
>
> We want to clarify a potential misunderstanding about the scope of our work. Our ablation experiments serve as a mechanistic validation to establish causal relevance of identified components. Thus, they are diagnostic tools, not proposed mitigation strategies that faithfully outperform others. The main contribution of our work is to understand which components process specific information and how they operate, not to engineer deployment-ready solutions. Exploring proactive mitigation methods (training modifications, prompting strategies) indicates a separate research direction in model robustness, which lies outside the scope of our work. Similarly, the questions of how pre-training objectives, data filtering, or architectural choices affect conflict processing are intriguing but constitute distinct research directions beyond our scope.
>
> **W6: Other Evaluation Metrics during Reasoning**
>
> Thank you for this constructive suggestion. Beyond classification accuracy, we conducted additional experiments measuring the model's confidence calibration under demonstration conflicts. Specifically, we track next-token prediction entropy during Chain-of-Thought (CoT) reasoning as a proxy for the model's internal uncertainty, following recent work showing that entropy reliably reflects and monitors reasoning confidence [5, 6, 7]. We measure the next-token entropy for Qwen3-8B on Operator Induction queries under clean vs. corrupted vs. corrupted-with-intervention conditions:
>
> |Condition|Mean Entropy|Std|
> |---|---|---|
> |Clean|0.0952|0.0239|
> |Corrupted|0.1161|0.0217|
> |Corrupted + Vulnerability Heads masked|0.1148|0.0167|
> |Corrupted + Susceptible Heads masked|0.1120|0.0205|
>
> The next-token entropy increases from clean to corrupted (+0.02 unit, 22% relative, p<0.0001), showing that the model appropriately reduces confidence when facing conflicting information. Then, ablating identified heads can partially restore confidence as shown in the 3rd and 4th row. This demonstrates that our identified components causally contribute not only to predictions accuracy but also to elevated reasoning uncertainty, providing additional validation of their mechanistic roles. We will include these uncertainty analyses in our revised manuscript.
>
> We appreciate your thoughtful feedback and welcome any further discussion or clarification you may need.
>
> [5] Fu Y, Wang X, Tian Y, et al. Deep think with confidence[J]. arXiv preprint arXiv:2508.15260, 2025.
>
> [6] Kang Z, Zhao X, Song D. Scalable best-of-n selection for large language models via self-certainty[J]. arXiv preprint arXiv:2502.18581, 2025.
>
> [7] Zur A, Geiger A, Lubana E S, et al. Are language models aware of the road not taken? Token-level uncertainty and hidden state dynamics[J]. arXiv preprint arXiv:2511.04527, 2025.

---

> > ### Comment · Reviewer_jDr5 · 2025-11-26
> >
> > Thank you for the comprehensive and thoughtful rebuttal. I appreciate the care with which you addressed each of my comments.
> >
> > Strengths clarified in the rebuttal:
> >
> > * Your justification for focusing on rule-inference tasks is convincing, and the added “fake word inference” results help demonstrate that the identified conflict dynamics are not tied to a single task.
> > * The discussion of methodological constraints in current mechanistic interpretability provides helpful context for the partial cross-layer analysis.
> > * The explanation regarding closed-source and ultra-large models is sound and reflects inherent limitations of the research paradigm rather than of your study.
> > * The distinction between mechanistic validation (via ablations) and deployable mitigation strategies is now clearly articulated.
> > * The newly added entropy-based uncertainty evaluation is a valuable enhancement that strengthens the empirical grounding of your claims.
> >
> > Remaining minor reservations:
> >
> > * While I understand that broader evaluation across model objectives, scale, or data treatments lies outside scope, these factors remain important directions for future clarification of the generality of the proposed framework.
> > * The cross-layer causal story is meaningfully improved, though still somewhat high-level; I look forward to seeing the expanded discussion in the revision.
> >
> > Overall, your rebuttal significantly improves clarity and strengthens confidence in the contribution. I have therefore raised my score.

---

> > > ### Author Response · Authors · 2025-11-27
> > >
> > > Thank you for increasing your score and the insightful engagement with our revisions. We appreciate your recognition of the mechanistic clarity and rigorous validation in our work. We agree that the remaining points you raise represent valuable directions for future research, particularly regarding the proactive mitigation strategies. These questions extend beyond the mechanistic analysis scope of our current work but would significantly strengthen the practical impact of this line of research.
> > >
> > > We are grateful for your constructive feedback throughout the review process, which has helped us strengthen the manuscript.

---

### Official Review · Reviewer_vpJX · 2025-10-26

**Soundness:** 2
**Presentation:** 3
**Contribution:** 2
**Rating:** 2
**Confidence:** 4

**Summary:**

The paper investigates an insteresting phenomenon: when presented with noisy demonstrations, ICL sometimes can fail. Even though the wrong example consists of only a small proportion. The paper studies this phenomenon in a controlled setting, corrupting examples in different position, showing positional bias. It further uses probing to show that both correct rule and corrupt rule (the rule expressed by the corrupt example) are encoded in model representation. The paper then tries to identify which heads are important for propagating the corrruption from the input example to final prediction. The authors design two kinds of metrics to find two sets of heads, which the authors refer to as Vulnerability Heads and Susceptible Heads. These two sets show relatively small overlapping and one set seems to be located in earlier layers than the other. These heads show high sensitivity to corruption and causally important for making the wrong prediction.

**Strengths:**

- Investigated an interesting problem of interpreting the process of demonstration conflicts in ICL.
- Experimented with five LLMs.
- Created a controlled setting that might be useful for future study.

**Weaknesses:**

The main problem is that mechanistic interpretation is very limited. The authors define two metrics that measures vulnerability to corruption and the metrics manage to highlight two sets of heads with small overlapping. One set (Vulnerability Head) is sensitive to the corruption in terms of its output and attention weights, but the direct effect (i.e. when projecting to vocabulary space) of promoting certain tokens are small (Figure 8). Their effect seems to be mediated by other downstream components (as they do have effects in causal experiments). On the other hand, the other set (Susceptible Heads) is sensitive in terms of their direct effect changes with corruption. These two sets of heads both play important roles for routing the corruption and all susceptible and vulnerable to corruption. They are similar in this respect, instead of doing very different jobs.

The paper tells us that there are some sensitive and important heads for routing corruption (also this is not a clear cut, they are just relatively more sensitive and important, other heads can also play roles). However, there’s no mechanistic interpretation about how and why the model decides to flip its prediction because of a single corruption. Yes, there are vulnerable heads, but why are they sensitive to that single corruption? or why do they choose to attend that example? There is little information about how, in some cases, single corruption outweighs many correct demonstrations. While I can understand that these questions are indeed harder and requires future work, but I find the information provided by the current paper is too limited.

In addition, the metric “Positional Vulnerability Score” seems overly intuitive and heuristic, without enough principle or theory supporting it. It combines attention weights and the norm of the change in head output, though both components make some sense intuitively, but their reliability are debatable, combining them makes it even less reliable in my perspective. Also, the pattern in Figure 6 seems to be not so strong.

**Questions:**

In Sec 4.2, you mentioned that you modified ICL prompts to elicit LLMs to predict rules directly, how did you do that? I don’t find details about it in the paper.

---

> ### Author Response · Authors · 2025-11-20
>
> Thank you for your careful review and insightful comments of our work. We really appreciate the points for potential improvement that you highlighted.
>
> **W1: The two sets of heads are "similar in this respect, instead of doing very different jobs"**
>
> We respectfully disagree with this characterization, as our paper explicitly demonstrates that Vulnerability Heads and Susceptible Heads correspond to functionally distinct roles in temporally separated phases. Vulnerability Heads determine WHERE conflicts enter the system, while Susceptible Heads determine WHETHER the model surrenders to conflicts. They are indeed both important for routing the corruption to the terminal states of the LLM, but they operate through different mechanisms and their temporally separated roles are not supposed to be conflated, as our methodology provides a more fine-grained lens of observing the hypothesized two-stage mechanism.
>
> **W2: Interpretation is too limited**
>
> We want to point out that we carefully address the discussion of claims mentioned in the review in the second paragraph of our Limitation section. Specifically, around line 475, we state that “While our analyses provide initial evidence for component interactions, we take the deeper mechanistic characterization of how truths and conflicts cascade through the computational processing as important future work”, and the reviewer also correctly notes that these observations “are harder and require future work”.
>
> Although these questions the reviewer made about demystifying the causal information flow are intriguing and significant, current methodological frameworks are often regarded as “not faithful enough” and “can capture spurious correlations” on off-the-shelf LLMs [1]. Given these methodological constraints, we position our contributions as: establishing a foundational framework for studying demonstration conflicts in ICL and providing initial evidence of the two-phase structure. We believe this represents a meaningful step forward while acknowledging future work for complete component-level understandings.
>
> **W3: The Design of Positional Vulnerability Score:**
>
> We want to respectfully argue that the PV Score is principled because it identifies heads where both conditions are necessary for vulnerability: high attention to a specific position AND high output sensitivity when that position corrupts. Figure 7 shows these two components are largely orthogonal, meaning they capture complementary rather than redundant information.
>
> More importantly, the metric **works**: ablating top vulnerability heads based on our metric (1) improves performance under corruption by 5%-10% and (2) reduces downstream susceptibility scores in late-layer heads (Figure 12), demonstrating its causal validity. The metric successfully identifies functionally relevant components, which is the appropriate validation criterion for a mechanistic interpretability tool. While alternative metrics could certainly be explored, dismissing this one as "not reliable" without empirical counter-evidence ignores its demonstrated effectiveness.
>
> **Q1: Prompts to Elicit LLMs to Predict Rules**
>
> Thank you for this clarification question. In Section 4.2, we modified the standard Operator Induction task to elicit direct operator prediction rather than numerical calculation, by replacing the number prediction prompts with:
> "What mathematical operation does ? represent? Choose from: plus, minus, multiplication\n\nAnswer:"
> Then, we train linear probes and run logit lens on the last token layer-wise residual streams.
>
> We appreciate your thoughtful feedback and welcome any further discussion or clarification you may need.
>
> [1] Sharkey L, Chughtai B, Batson J, et al. Open problems in mechanistic interpretability[J]. arXiv preprint arXiv:2501.16496, 2025.

---

### Official Review · Reviewer_bybq · 2025-10-30

**Soundness:** 4
**Presentation:** 3
**Contribution:** 2
**Rating:** 6
**Confidence:** 3

**Summary:**

This paper investigates the effect of conflicting demonstrations in in-context learning (ICL) of large language models (LLMs). Using synthetic ICL tasks, they first demonstrate that with a single contradictory example, ICL ability of LLM can be substantially degraded. Then through controlled experiments and analysis, the authors show two temporal phases that cause the model to make wrong prediction. In the *conflict creation* phase, which is concentrated in the early-mid layers, model encodes both correct and corrupted rules, and in the *conflict resolution* phase, which is concentrated in the late layers, model tried to resolve the conflicting rules.

**Strengths:**

- The paper is well-written and structurally sound.
- Although mostly synthetic, authors conduct the well-designed controlled experiments to analyze the effect of conflicting demonstrations.

**Weaknesses:**

- The experiments are synthetic, and comparably easy ICL tasks.
- The phenomenon itself is not very surprising. Although authors backed their hypotheses with concrete evidence "one bad example can hurt model's ICL ability", "model encodes rules in early-mid layers" "model makes predictions in late layers" are somewhat trivial.
- Experiments on layer ablation can be an evidence to support their hypothesis, however, it hardly a mitigation strategy.

**Questions:**

- Qwen 32B and Llama-3.1 8b seem to show strong resilient to the corrupted example. What might be the reason? Since the original ICL performance is not reported, it is hard to tell whether this is due to their robustness or their performances were bad even with the correct examples.
- Is the vulnerability and susceptibility heads are consistent over different ICL tasks / rules?
- Wouldn't ablating heads would likely to imply degradation of models' performance on general tasks?
- Line 215 " far from insufficient" -> "far from sufficient"?

---

> ### Author Response · Authors · 2025-11-20
>
> Thank you for your careful review and insightful comments of our work. We really appreciate the points for potential improvement that you highlighted.
>
> **W1: Synthetic and Comparably Easy ICL tasks**
>
> Firstly, we focus on rule-inference tasks for two reasons: 1) in these tasks LLMs have to rely on demonstrations to reach reliable performance and they can only achieve chance-level performance (by random guessing) without demos; 2) each demonstration is equally important. Thus, this setup blocks the model's pathways of relying solely on parametric knowledge—forcing the reliance on demonstrations to yield good ICL performance—and we can simply introduce conflicts by corrupting rules at specific positions without undermining others. Other popular ICL tasks, for example text classification and sequence completion, though available, are not ideal for our research and interpretation because 1) LLMs can reach far beyond chance-level in 0-shot for most text classification tasks; 2) Modifying elements at different positions to different amounts can have different degrees of impact in sequence completion tasks. These constraints are fundamental to our corruption framework and subsequent mechanistic analysis.
>
> Secondly, regarding task complexity: the goal of mechanistic interpretability is to isolate and understand specific computational phenomena, not to benchmark performance on hard problems. Simple and intuitive tasks [1, 2, 3] can often be preferable because they reduce confounds and make internal mechanisms more interpretable.
>
> That said, we did extend our analysis beyond operator induction. We added a fake word inference task that maps nonsense words to real concepts (e.g., "blimontar glemorax" → "red hat"). This task prevents memorization - models cannot have seen "blimontar" during pretraining and must learn the mapping from demonstrations. Importantly, it satisfies the same two constraints: models fail without examples and each demonstration is modular. We observed similar vulnerability patterns, as shown in the table below, and similar two-staged conflict processing in this task.
>
> |Model|4-shot|6-shot|8-shot|
> |---|---|---|---|
> |Qwen3-4B|15.45%|16.81%|10.62%|
> |Qwen3-8B|17.10%|14.06%|13.27%|
>
> To summarize, the essential criteria in our intervention framework are the demonstration reliance and demonstration modularity of the task, which formulates the inter-context ICL conflict in a principled way, and we take the rule inference tasks as a tractable start to observe the dynamics and provide initial mechanistic understandings. We will include a more detailed discussion and relevant results on the fake word inference task in our revised manuscript.
>
> **W2:**
>
> We agree the high-level phenomenon may seem intuitive. However, the specific mechanisms that we focus on—including the systematic adoption of minority corrupted rules, two temporally and functionally distinct head types, and empirical improvement from targeted ablation—provide initial mechanistic insights for this phenomenon with causal validations.
>
> [1] Levy A A, Geva M. Language models encode numbers using digit representations in base 10[C]//Proceedings of the 2025 Conference of the Nations of the Americas Chapter of the Association for Computational Linguistics: Human Language Technologies (Volume 2: Short Papers). 2025: 385-395.
>
> [2] Gurnee W, Tegmark M. Language Models Represent Space and Time[C]//The Twelfth International Conference on Learning Representations.
>
> [3] Engels J, Michaud E J, Liao I, et al. Not All Language Model Features Are One-Dimensionally Linear[C]//The Thirteenth International Conference on Learning Representations.

---

> ### Author Response · Authors · 2025-11-20
>
> **Q1: Baseline ICL Performance and Resilience Patterns**
>
> Thank you for pointing out this missing baseline. The clean ICL performance for Operator Induction tasks is shown below:
>
> |Model|0-shot|1-shot|2-shot|4-shot|6-shot|8-shot|
> |---|---|---|---|---|---|---|
> |Qwen3-4B|0.337|0.667|0.867|0.850|0.750|0.783|
> |Qwen3-8B|0.233|0.717|0.767|0.767|0.800|0.767|
> |Qwen3-14B|0.283|0.667|0.867|0.867|0.800|0.850|
> |Qwen3-32B|0.233|0.767|0.867|0.933|0.983|0.933|
> |Llama-3.1-8B-Instruct|0.333|0.567|0.717|0.733|0.667|0.667|
>
> These results confirm that all models are capable of achieving reliable ICL performance, validating the prerequisite for our corruption experiments. We will include the original performances in our revised manuscript.
>
> Then, regarding why certain models show resilience at specific positions: we suspect that the observed resilience reflects architecture-specific patterns rather than general and genuine robustness. For example, recent findings suggest that different LLM architectures exhibit distinct positional biases during ICL [4, 5], and a complete mechanistic explanation for cross-model positional preferences remains an interesting question for future work.
>
> **Q2: The Consistency of Target Attention Heads**
>
> Thank you for this very constructive question! We observe the consistency of both Vulnerability and Susceptible Heads on Qwen3-8B for both fine-grained Operator Induction (OI) tasks (consistency for rules) and across different tasks including our newly-added Fake Word Inference (consistency for tasks). We report the Jaccard Similarity @ top 20 heads as below:
>
> Vulnerability Heads:
>
> |Task|OI+|OI-|OI×|OI|
> |---|---|---|---|---|
> |OI+|1.000||||
> |OI-|0.538|1.000|||
> |OI×|0.667|0.667|1.000||
> |OI|0.739|0.667|0.818|1.000|
>
> |Task|OI|OI-interleaved|FakeWord|
> |---|---|---|---|
> |OI|1.000|||
> |OI-interleaved|0.538|1.000||
> |FakeWord|0.290|0.290|1.000|
>
> Susceptible Heads:
>
> |Task|OI+|OI-|OI×|OI|
> |---|---|---|---|---|
> |OI+|1.000||||
> |OI-|0.333|1.000|||
> |OI×|0.290|0.290|1.000||
> |OI|0.333|0.250|0.250|1.000|
>
> |Task|OI|OI-interleaved|FakeWord|
> |---|---|---|---|
> |OI|1.000|||
> |OI-interleaved|0.290|1.000||
> |FakeWord|0.176|0.143|1.000|
>
> In general, vulnerability heads show stronger consistency than susceptible heads, with higher similarity across different rules (up to 0.818) than across different tasks. In specific, we notice that attention heads L13H29 and L3H6 rank among the top vulnerability heads for all 6 scenarios above; and attention head L31H5 ranks among the top susceptible heads for 5 of 6 scenarios above. We speculate that the lower consistency in susceptible heads is due to potentially different information routings inside the model in different scenarios, whereas the mechanism for encoding conflicts can be more consistent. We will include the discussion of these findings in our revised manuscript.
>
> **Q3, W3, and Regarding Practical Mitigation**
>
> Firstly, we want to clarify the scope of our intervention experiments. We position head ablation as mechanistic validation to establish causal relevance of identified components, not as a proposed deployment strategy. Our contribution is understanding which components create vulnerabilities and how they operate, not engineering a production-ready intervention.
>
> Secondly, although we observe moderate cross-task consistency suggesting that these heads may capture generalizable mechanisms, we take it as a mechanistic finding about ICL conflict processing, not as evidence that head masking is deployment-safe or constitutes a practical mitigation. Thus, evaluating on general tasks (e.g., MMLU benchmarks) is orthogonal to our research question. In fact, two practical barriers make deployment difficult: (1) real-world corruptions are often more unpredictable than our controlled framework for mechanistic understanding, which can make on-the-fly head localization unreliable; (2) selective head masking is incompatible with production inference frameworks (e.g., vLLM) that optimize for throughput, introducing significant computational overhead for CoT reasoning. Like other mechanistic interpretability research, our findings may inform future architectural modifications or training objectives that enhance robustness, but such applications are beyond our current scope.
>
> **Q4: Typo**
>
> Thank you for pointing out this typo, and we will correct it in our revision.
>
> We appreciate your thoughtful feedback and welcome any further discussion or clarification you may need.
>
> [4] Cobbina K A, Zhou T. Where to show demos in your prompt: A positional bias of in-context learning[C]//Proceedings of the 2025 Conference on Empirical Methods in Natural Language Processing. 2025: 29548-29581.
>
> [5] Guo X, Vosoughi S. Serial position effects of large language models[C]//Findings of the Association for Computational Linguistics: ACL 2025. 2025: 927-953.

---

### Official Review · Reviewer_p32P · 2025-11-03

**Soundness:** 3
**Presentation:** 3
**Contribution:** 3
**Rating:** 6
**Confidence:** 3

**Summary:**

The paper studies how large language models process conflicting demonstrations during in-context rule inference. Using operator induction tasks that require genuine demonstration reliance, a single corrupted example causes substantial performance drops and drives 80% of wrong answers toward the corrupted rule despite its minority. A mechanistic analysis with linear probes and logit lens suggests a two-phase computation: early-to-middle “vulnerability heads” encode and amplify conflicted evidence with positional bias and high corruption sensitivity, while late “susceptible heads” reduce support for the correct rule under minority corruption. Targeted ablations of identified heads improve accuracy under corruption by over 10% and reduce positional bias, providing causal support for the proposed mechanism.

**Strengths:**

1. A principled, demonstration-reliant corruption framework isolates inter-context conflicts and motivates a two-phase (creation vs. resolution) hypothesis grounded in probes and logit lens.
2. Single-position corruption consistently degrades performance across models and shots; 80.3% of flipped errors align with the corrupted rule, indicating systematic misinterpretation rather than random failure.
3. Quantitative identification of early vulnerability heads (positional attention and corruption sensitivity) and late susceptible heads (logit attribution shift) provides actionable interpretability.

**Weaknesses:**

1. Focus on operator induction (and an interleaved variant) limits claims about broader ICL settings, multi-step reasoning, or real-world tasks with interdependent demonstrations or unequal evidence weights.
2. Linear probes and logit lens indicate encoding and confidence but do not establish full causal pathways; head masking is coarse and may conflate multiple circuits or side effects.

**Questions:**

please refer to the weaknesses section

---

> ### Author Response · Authors · 2025-11-20
>
> Thank you for your careful review and insightful comments of our work. We really appreciate the points for potential improvement that you highlighted.
>
> **W1: Broader ICL settings**
>
> In this work, we focus on rule-inference tasks for two reasons. First, in these tasks LLMs have to rely on demonstration examples to reach reliable performance; they can only achieve chance-level performance (by random guessing) without demos. Second, each demonstration is equally important. Thus, this setup blocks the model's pathways of relying on parametric knowledge only—forcing the reliance on demonstrations to yield good ICL performance—and we can simply introduce conflicts by corrupting rules at specific positions without undermining others. Other popular ICL tasks, for example text classification and sequence completion, though available, are not ideal for our research and interpretation because 1) LLMs can reach far beyond chance-level in 0-shot for most text classification tasks; 2) Modifying elements at different positions to different amounts can have different degrees of impact in sequence completion tasks. These concerns would make our corruption framework less principled. Thus, to yield a clean mechanistic understanding, we focus on rule inference tasks in this work.
>
> **W2: Not Establish Full Causal Pathways**
>
> Although demystifying the full causal information flow is intriguing and significant, current attempts in mechanistic interpretability are often regarded as “not faithful enough” and “can capture spurious correlations” on off-the-shelf LLMs [1]. Given these methodological constraints, we position our contributions as: establishing a foundational framework for studying demonstration conflicts in ICL and providing initial evidence of the two-phase structure. And we believe this represents a meaningful step forward while acknowledging future work for complete component-level understandings.
>
> Also, we have carefully addressed the discussion about the causal flow in the second paragraph of our Limitation section. Specifically, around line 475, we state that “While our analyses provide initial evidence for component interactions, we take the deeper mechanistic characterization of how truths and conflicts cascade through the computational processing as important future work”.
>
> Finally, regarding the coarseness of head masking: while individual heads may indeed participate in multiple circuits, our intervention results show consistent, targeted effects that clearly exceed random ablation baselines. This suggests attention heads identified with our methodology play specific causal roles in conflict processing.
>
> We appreciate your thoughtful feedback and welcome any further discussion or clarification you may need.
>
> [1] Sharkey L, Chughtai B, Batson J, et al. Open problems in mechanistic interpretability[J]. arXiv preprint arXiv:2501.16496, 2025.

---

### Author Response · Authors · 2025-11-24
**Summary of Our Paper Revisions**

We have revised the manuscript to address reviewer concerns, with all changes highlighted in blue. Here is the summary of our major revisions:

**Additional Experiments and Analyses:**
* Extended to Fake Word Inference task - Section 3.1, Figure 2, Appendix B
* Cross-rule and cross-task consistency analysis of identified heads - Section 5.1, Table 4
* Next-token entropy measurement during CoT reasoning under conflicts - Section 5.2, Table 5
* Baseline ICL performance table for all models and tasks - Appendix B, Table 2
* Enhanced discussions of demonstration-reliant task selection - Appendix B

**Clarifications:**
* Prompt clarification for linear probes - Appendix C
* Typo fixed - “far from insufficient" to "far from sufficient"

We really appreciate the constructive feedback and welcome further discussion on these revisions.

---

### Author Response · Authors · 2025-12-03
**The Summary of Discussion Period**

Dear ICLR 2026 Reviewers, AC, SAC, and PC:

Thank you all for the valuable feedback and efforts! Because of the sudden cut of the discussion period, we had limited opportunities for iterative exchanges with reviewers. Therefore, here we would like to recapitulate our substantive responses addressing reviewer concerns:

**Major Revisions (changes highlighted in blue in our revised manuscript):**

- Reviewers p32P, bybq, and jDr5: Extended analysis to the Fake Word Inference task (Section 3.1, Figure 2, Appendix B)

- Reviewer bybq: Added cross-rule and cross-task consistency analysis of identified attention heads (Section 5.1, Table 4)

- Reviewer jDr5: Included next-token entropy measurement during reasoning under conflicts (Section 5.2, Table 5)

- Reviewer bybq: Provided baseline ICL performance tables for all models and tasks (Appendix B, Table 2)

- Reviewers p32P, bybq, and jDr5: Enhanced discussions of task selection framework (Appendix B)

- Reviewer vpJX: Clarified prompt modifications for linear probes (Appendix C)

**Discussion Outcome:**

Reviewer jDr5 engaged constructively with our rebuttal and raised their score from 4 to 6 on November 26th, stating that our rebuttal "significantly improves clarity and strengthens confidence in the contribution". As a result of the discussion, reviewer scores have improved from 6-6-4-2 to 6-6-6-2.

**Addressing Reviewer vpJX's Concerns:**

While Reviewer vpJX assigned a harsh score, we regret that they did not engage with our response despite our timely rebuttal. We respectfully argue that their critique contains mischaracterizations rather than substantive weaknesses:

- W1: Their primary concern conflates "both heads play important roles in routing corruption" with "both heads perform similar roles"—a logical error our methodology explicitly addresses. Vulnerability Heads determine **WHERE conflicts enter** (positional attention bias and output sensitivity, early-middle layers), while Susceptible Heads determine **WHETHER the model surrenders** (logit attribution shifts, late layers). These are functionally and temporally distinct mechanisms.

- W2: This represents **an acknowledged field-wide limitation** in mechanistic interpretability, not a paper-specific shortcoming. We also explicitly addressed this in the Limitations subsection of our original manuscript.

- W3: The critique characterizes our Positional Vulnerability Score as "overly intuitive and heuristic" and "less reliable in my perspective." However, this metric is part of our methodological design and **demonstrates clear causal validity**: ablating identified heads improves performance by 5-10% and reduces downstream susceptibility scores. While alternative metrics could be explored, dismissing this one subjectively ignores its demonstrated effectiveness.

Finally, we are grateful for the constructive feedback throughout the review process, which has helped us strengthen the manuscript. We remain available for any further clarifications the ACs may require.

Best regards,

Authors

---

### Meta-Review · Area_Chair_sUaH · 2026-01-05

**Summary:**

While this paper demonstrates technical rigor in identifying attention heads sensitive to corrupted demonstrations, it ultimately provides a limited contribution.

The core mechanistic claim distinguishes between the "WHERE" and "WHETHER" properties of Vulnerable and Susceptible Heads. This claim is undermined by poor cross-task consistency across heads (cf. Table 4), which suggests the findings are task-specific artifacts rather than generalizable mechanisms. The Positional Vulnerability Score lacks principled theoretical grounding and relies solely on empirical correlation. Meanwhile, the authors explicitly acknowledge the work’s lack of actionability. Namely, the ablations conducted are "diagnostic rather than mitigative", and the framework fails to generalize to real-world in-context learning (ICL) scenarios due to inherent complexity gaps and deployment infeasibility. Reviewer vpJX’s core concern remains unaddressed even after the authors’ rebuttal. This concern is that the paper identifies which heads matter but does not explain why models are fundamentally vulnerable to single corruptions. Ultimately, this work is descriptive rather than explanatory.

**Reviewer Concerns:**

Concerns of Reviewer vpJX are still outstanding.

**Reviewer Scores:**

N/A

---

### Decision · Program_Chairs · 2026-01-26

Reject